# Deep sequencing of DNA from urine of kidney allograft recipients to estimate donor/recipient-specific DNA fractions

**Aziz Belkadi**[1], **Gaurav Thareja**[1], **Darshana Dadhania**[2,3], **John R. Lee**[2,3], **Thangamani Muthukumar**[2,3], **Catherine Snopkowski**[3], **Carol Li**[3], **Anna Halama**[1], **Sara Abdelkader**[1], **Silvana Abdulla**[1], **Yasmin Mahmoud**[1], **Joel Malek**[1], **Manikkam Suthanthiran**[2,3], **Karsten Suhre**[1] *

**1** Department of Physiology and Biophysics, Weill Cornell Medicine-Qatar, Education City, Doha, Qatar, **2** Department of Transplantation Medicine, New-York Presbyterian Hospital-Weill Cornell Medicine, New York, United States of America, **3** Division of Nephrology and Hypertension, Department of Medicine, Weill Cornell Medical College, New York, United States of America

\* kas2049@qatar-med.cornell.edu

**Data Availability Statement:** Data cannot be shared publicly because the Weill Cornell Medicine ethics committee has imposed restrictions on sharing a de-identified data set. Data are available

## Abstract

Kidney transplantation is the treatment of choice for patients with end-stage kidney failure, but transplanted allograft could be affected by viral and bacterial infections and by immune rejection. The standard test for the diagnosis of acute pathologies in kidney transplants is kidney biopsy. However, noninvasive tests would be desirable. Various methods using different techniques have been developed by the transplantation community. But these methods require improvements. We present here a cost-effective method for kidney rejection diagnosis that estimates donor/recipient-specific DNA fraction in recipient urine by sequencing urinary cell DNA. We hypothesized that in the no-pathology stage, the largest tissue types present in recipient urine are donor kidney cells, and in case of rejection, a larger number of recipient immune cells would be observed. Extensive in-silico simulation was used to tune the sequencing parameters: number of variants and depth of coverage. Sequencing of DNA mixture from 2 healthy individuals showed the method is highly predictive (maximum error < 0.04). We then demonstrated the insignificant impact of familial relationship and ethnicity using an in-house and public database. Lastly, we performed deep DNA sequencing of urinary cell pellets from 32 biopsy-matched samples representing two pathology groups: acute rejection (AR, 11 samples) and acute tubular injury (ATI, 12 samples) and 9 samples with no pathology. We found a significant association between the donor/recipient-specific DNA fraction in the two pathology groups compared to no pathology (P = 0.0064 for AR and P = 0.026 for ATI). We conclude that deep DNA sequencing of urinary cells from kidney allograft recipients offers a noninvasive means of diagnosing acute pathologies in the human kidney allograft.

from the Weill Cornell Medicine Institutional Data Access / Ethics Committee (contact via irb@med. cornell.edu) for researchers who meet the criteria for access to confidential data. The data underlying the results presented in the study are available from irb@med.cornell.edu.

**Funding:** This work was supported by the Biomedical Research Program at Weill Cornell Medicine–Qatar, a program funded by the Qatar Foundation (https://www.qf.org.qa) [to AB, GT, AH, Sa A, Si A, Y M, J M and K S], the National Priority Research program of the Qatar National research fund (https://www.qnrf.org/en-us/) [grant # NPRP12S-0227-190173 to AB and MS], the National Institute of Health (https://www.nih.gov) [grants # K08DK087824 and R37AI051652 to MS], The funders had no role in study design, data collection and analysis, decision to publish, or preparation of the manuscript.

**Competing interests:** The authors have declared that no competing interests exist.

# Introduction

In 1933 Ukrainian surgeon Yurii Voronoy achieved the first human kidney transplantation [1]. Kidney transplantation is the preferred treatment option for patients with end-stage renal failure compared to dialysis. Today, renal transplantation plays an important role in clinical medicine and has become a relatively safe intervention. However, various pathologies can still affect the transplanted organ, including infections, disease recurrence and immune rejections The rejections can be related to a range of donor- and recipient-specific factors [2, 3]. Acute renal rejection affects 10 to 20% of transplants within three months after transplantation and chronic rejection is an important cause of graft failure [3–5].

To diagnose rejection, kidney allograft biopsies are considered the gold-standard for detecting both acute and chronic immune rejection, as well as other associated pathologies that may eventually lead to allograft loss. However, biopsies are invasive, costly, and in rare cases can lead to organ loss, while the readout can potentially be erroneous if a non-affected part of the kidney is sampled by chance. Therefore, invasive biopsies in patients with low immunological risk could be criticized [6]. There is hence a strong need for noninvasive assays to detect injury in transplanted kidneys. Several studies to develop suitable biomarkers for allograft rejection have been conducted. These studies include the quantification of specific messenger RNAs (mRNA) in urine [7], large-scale transcriptomics analyses of peripheral blood [8], proteomics analyses of biopsies [9] and urine [10, 11], and metabolomics [12] and RNA sequencing [13] of urine pellet or supernatant. Nevertheless, non-invasive methods developed to date still have important caveats and require further improvement.

The presence of donor-specific DNA in the blood was first reported in women who had a kidney or a liver transplant [14]. The measurement of cell-free donor-specific DNA in blood for a differential diagnosis of kidney injury has been suggested recently [15–17]. These studies focused on females who received a kidney from male donors by identifying the presence of DNA coding for the testis specific protein *Y- linked 1* (TSPY1) or the sex-determining region of the Y chromosome using quantitative polymerase chain reaction. With the improvement of next generation sequencing technologies, whole genome sequencing (WGS) [18, 19] and targeted sequencing [20] were used for measuring donor-specific DNA for solid organ transplant rejection. However, these studies focused on heart transplants and measured cell-free donor-specific DNA in blood plasma. More importantly, these methods require the sequencing of both donor and receptor DNA which is more costly.

An algorithm for measuring donor-specific DNA in plasma of organ transplants without requiring donor or recipient genotyping was implemented by Gordon et al [21]. But this algorithm made the assumption that donor fraction is < 14%. More recently, Grskovic *et al.* used sequencing of 266 single nucleotide variants (SNVs) that discriminate best between two unrelated individuals to count reference and alternative allele frequency for estimating the donor-derived cell-free DNA fraction [22]. This method showed a high correlation between cell-free donor specific DNA levels in recipient blood and active rejection of the kidney allografts [23]. However, this method does not account for potential sequencing errors and requires *a priori* knowledge of the familial relationship between donor and recipient. Finally, a statistical method combining SNV array genotyping of donor and recipient before transplantation with recipient DNA sequencing was used to estimate recipient-derived DNA fraction in heart and lung transplants [24]. Nonetheless, this method requires SNV genotyping of donor and recipient DNA before transplantation. Most importantly, all of the previous studies focused on DNA extracts from blood.

The presence of donor-specific DNA in urine of kidney allograft recipients has been reported [25]. We recently conducted a study based on RNA sequencing of kidney allograft

biopsies and found a correlation between the ratio of heterozygous to homozygous SNVs with the rejection phenotype [26]. Moreover, we have shown that DNA methylation could be used to accurately estimate the tissue type composition in recipient urine samples. We found that the largest tissue types present in recipient urine were kidney cells and neutrophils and that donor-specific DNA fraction correlates with the kidney derived cell fraction [27]. However, we restricted the analysis to kidney recipients with urinary tract infection and BK-virus nephropathy only. Most recently, we have identified different gene signatures and pathways associated with two different types of kidney rejection using RNA-seq of urinary cell pellets: acute T cell–mediated rejection and antibody-mediated rejection [13]. Deconvolution analysis showed a higher enrichment of immune cells in urine matched to rejection biopsies compared to urine matched to no-rejection biopsies.

Given that the fraction of donor-specific DNA can be determined using DNA sequencing, we here hypothesize that the recipient-specific DNA fraction in urine correlates with the level of active rejection in the kidney allograft, assuming that recipient-specific DNA originates mostly from allograft-invading immune cells while donor-specific DNA comes from the allograft [28]. Inspired by methods to estimate DNA contamination in sequencing projects [29, 30], we present a cost-effective method to determine the fraction of donor/recipient-specific DNA (denoted $\alpha$ hereafter) in urine by sequencing targeted regions. We demonstrate that the precision of this measure depends on sequencing depth and length of the targeted region. Most importantly, no prior knowledge of donor and recipient relationship is required. To the best of our knowledge, this is the first method for estimating donor/recipient-specific DNA fraction in DNA mixture extracted from kidney graft recipient's urine. Our method provides an easy way to determine the donor/recipient-specific DNA fraction regardless of donor and recipient gender. We evaluate the applicability of our approach for the detection of kidney transplant rejection. Future applications could be routine tests of urine samples as a reference to adjust and personalize the dosage of immune suppressive drugs in kidney transplant patients.

## Materials and methods

### Algorithm

The proposed algorithm is inspired by the contamination estimation assessment in DNA sequencing methods [29, 30]. We hypothesize that recipient urine contains a mixture of recipient and donor DNA. Let $N$ be the number of bi-allelic SNVs sequenced from recipient urine DNA and each SNV $i$ is covered by $M_i$ reads. Let $g_i^R$ and $g_i^D$ be the genotype of recipient and donor at the SNV $i$, respectively. Both $g_i^R$ and $g_i^D$ are unknown. Limiting the analysis on bi-allelic SNVs only leads to three possible genotypes for recipient and donor at each SNV $i$: $g_i^R$ ($g_i^D$) = {0, 1, 2} where 0 = homozygous wild type, 1 = heterozygous and 2 = homozygous for the alternative allele. The likelihood of the donor-specific DNA fraction ($\alpha$) will be:

$$L(\alpha) = \prod_{i=1}^{N} \sum_{g_i^R} \sum_{g_i^D} \left\{ \prod_{j=1}^{M_i} \sum_{e_{ij}} \left( (1-\alpha) P\left(b_{ij}|g_i^R, e_{ij}\right) + \alpha P\left(b_{ij}|g_i^D, e_{ij}\right) \right) P(e_{ij}) \right\} P(g_i^R) P(g_i^D) \quad (1)$$

Where $b_{ij}$ represents the read $j$ covering the SNV $i$. $e_{ij}$ represents the sequencing error of SNV $i$ at the read $j$: $P(e_{ij} = 1) = 10^{-Q_{ij}/10}$ and $P(e_{ij} = 0) = 1 - P(e_{ij} = 1)$. $Q_{ij}$ represents the minimum between the base quality of the read $j$ at the position of the variant $i$ and the mapping quality of the read $j$. The probability of $b_{ij}$ conditioned to the recipient (donor) genotype $g_i^R$ ($g_i^D$) and the sequencing error $e_{ij}$ is described in Table 1. Finally, we used the simulated annealing approach together with a grid search to find $\alpha$ that maximizes the likelihood function [31]. The method was implemented in a Python script.

**Table 1. The probability of read $b_{ij}$ carrying the reference (a), alternative (A) or a different allele (e) conditioned to the recipient (donor) genotype $g_i^R$ ($g_i^D$) and the sequencing error $e_{ij}$.**

|  | $g_i^R$ ($g_i^D$) = 0 | | $g_i^R$ ($g_i^D$) = 1 | | $g_i^R$ ($g_i^D$) = 2 | |
|---|---|---|---|---|---|---|
|  | $e_{ij} = 0$ | $e_{ij} = 1$ | $e_{ij} = 0$ | $e_{ij} = 1$ | $e_{ij} = 0$ | $e_{ij} = 1$ |
| P($b_{ij}$ = a) | 1 | 0 | 1/2 | 1/6 | 0 | 1/3 |
| P($b_{ij}$ = A) | 0 | 1/3 | 1/2 | 1/6 | 1 | 0 |
| P($b_{ij}$ = e) | 0 | 2/3 | 0 | 2/3 | 0 | 2/3 |

As the likelihood function for estimating α requires a balance in alternative/reference allele distribution in heterozygous calls for both recipient $g_i^R$ and donor $g_i^D$ genotypes, very deep recipient urine DNA sequencing will provide this allele balance (Table 2).

## Simulated SNVs based on general population structure

To assess the effect of the number of SNVs (*N*) and mean depth of coverage (*M*) on the allele balance and thus the prediction accuracy of the likelihood function, we simulated 2 independent SNV-sets each set containing *N* common SNVs (minor allele frequency ≥ 5%) and covered on average by *M* reads. We varied *N* and *M* in 35 scenarios where *N* = {10, 50, 100, 500 and 1,000} and *M* = {10, 50, 100, 500, 1,000, 5,000 and 10,000}. We merged α reads from set1 and 1-α from set2 randomly generating a combined SNV-set and applied the likelihood function on the combined SNV-set to estimate the *observed α*. Because *L(α) = L(1-α)*, we restricted $0 \leq \alpha \leq 0.5$ in steps of 0.01 generating 51 scenarios. A thousand replicates for each scenario and for each α were performed to obtain an empirical distribution.

## Sequencing of urinary cell DNA from a pair of healthy individuals

We extracted DNA from whole urinary cell pellet of two healthy individuals; S1, a 30-year-old European woman and S2 a 30-year-old Arab woman using the Qiagen® Allprep Mini Kit (S2 Table in S1 File) extraction kit. We omitted urinary cell-free DNA as it has been shown that urinary fragmented DNA needs suitable preservatives to avoid DNA degradation [32]. The DNA concentration was similar for the two individuals: 35ng/µl. We mixed DNA from S1 and S2 to achieve 5 scenarios: i) 100% from S1; ii) 100% from S2; iii) 90% from S1; and 10% from S2; iv) 70% from S1 and 30% from S2; v) 50% from S1 and 50% from S2, and each scenario was replicated three times. We performed deep targeted DNA sequencing on each replicate. GeneReadDNA Seq Targeted Panels V2; Human Breast Cancer Panel (Qiagen, USA) was used to perform target enrichment by multiplex PCR. The breast cancer panel consists of four primer pools yielding 2,915 amplicons. Briefly, 40ng of each gDNAs was amplified using PCR reagents with 4 primer pool mixes following the manufacturer's protocol. After the completion of the 4

**Table 2. The probability and the 99% interval confidence of a perfect allele balance in a heterozygous call as a function of depth of coverage *M*.** A total of 10,000 simulations were performed for each proposed *M*.

| M | P(alt/(ref+alt) = 0.5) | 99% confidence interval |
|---|---|---|
| 10 | 0.24 | [0.10, 0.90] |
| 50 | 0.11 | [0.34, 0.66] |
| 100 | 0.08 | [0.39, 0.62] |
| 500 | 0.18 | [0.45, 0.55] |
| 1,000 | 0.27 | [0.46, 0.54] |
| 5,000 | 0.53 | [0.48, 0.52] |
| 10,000 | 0.69 | [0.49, 0.51] |

PCR reactions, the 4 products were combined, and the enriched DNA was purified using Agencourt AMPure XP beads (Beckman Coulter, USA). The concentration and the size of the purified amplicons were determined using Qubit 2.0 Fluerometer dsDNA BR assay kit (Life-Technologies, USA) and Agilent BioAnalyzer 2100 High-Sensitivity DNA kit (Agilent Technologies, USA). A total amount of 80–160 ng of purified enriched DNA was used as template to generate NGS libraries. The NGS libraries were prepared using NEBNEXT Ultra II DNA Library Prep Kit (New England Biolabs, USA) and NEXTflex DNA Barcodes (Bio Scientific, USA). All library preparation steeps were performed according to the manufacturer's protocol. The size and quality of the final libraries were analyzed using Agilent BioAnalyzer 2100 with 1000 DNA kit (Agilent Technologies, USA). The quantified libraries were then normalized, pooled, and spiked with 5% PhiX control library (Illumina, USA). Finally, the pooled libraries were sequenced on a single lane of Illumina Hiseq 4000 (Illumina, USA) paired-end 150 bp run.

Obtained reads were aligned to the human genome reference hg19 using bwa [33]. A total of 51,893 bi-allelic SNVs from the Exac project are included in the targeted genomic regions. The method works only on SNVs with different genotypes between donor and recipient. Under Hardy Weinberg assumption, we assessed the probability of having a different genotype for each SNV $i$ as:

$$P\left(G_i^D \neq G_i^R\right) = \sum_{G^D=0}^{2} \sum_{G^R=0}^{2} F\left(G_i^D = G^D\right) * F\left(G_i^R = G^R\right), G^D \neq G^R \tag{2}$$

$$F\left(G_i^{D(R)} = 0\right) = p_i^{D(R)^2} \tag{3}$$

$$F\left(G_i^{D(R)} = 1\right) = 2 * p_i^{D(R)} * q_i^{D(R)} \tag{4}$$

$$F\left(G_i^{D(R)} = 2\right) = q_i^{D(R)^2} \tag{5}$$

Where $G_i^D$ and $G_i^R$ represent the donor and recipient genotype, respectively. $p_i^D$ and $p_i^R$ represent the donor and recipient reference allele frequency, respectively. $q_i^D$ and $q_i^R$ represent the donor and recipient alternative allele frequency, respectively.

To avoid allele dropout due to primer annealing region, we filtered out 24,237 SNVs falling at the primer sequencing regions and SNVs carried by reads targeted by primers containing SNVs [34]. From the 27,656 remaining SNVs, we selected the 1,000 most common and applied the likelihood function after filtering out the reads carrying the variant at the last 20 base pairs [35].

## Simulated SNVs in pairs of individuals from the same and different ethnicities

We used individuals from the 1,000 Genomes Project phase 3 representing five major populations: AFR, AMR, EAS, EUR and SAS [36]. We randomly selected two individuals and aimed to cover all possible situations: five cases where individual1 and individual 2 belong to the same population and ten cases where individual1 and individual 2 belong to different populations. We extracted the 1,000 SNVs described previously from individual1 and Individual 2 and then merged α reads from individual1 and 1-α from individual 2 generating a combined SNV-set. We varied α from 0 to 0.5 in steps of 0.01 and fixed the mean depth of coverage at $M = 5,000$. We applied the likelihood function to assess α for each combined SNV-set generated. We repeated the individual selecting process 100 times to obtain an empirical distribution for each situation.

## Simulated SNVs in pairs of biological siblings

We performed WGS on Illumina HiSeq 2500 sequencer of 91 Qatari siblings from 27 nuclear families containing at least 2 siblings and up to 7 siblings [37]. Reads were aligned to the hg19 reference genome using bwa [33]. Sequence alignment files were filtered and genotypes were called using the Genome Analysis Tool Kit best practices pipeline; variants were called using HaplotypeCaller [38, 39]. We used Plink identity by state to confirm the familial relationship [40] (S1 Fig in S1 File). We extracted the 1,000 SNVs described previously from each sibling and merged each pair generating 100 combined SNV-sets. We applied the likelihood function to assess α for each combined SNV-set generated by varying α from 0 to 0.5 with a step of 0.01 while the mean depth of coverage was set at $M = 5,000$.

## Donor/recipient-specific DNA fraction in real kidney recipient urine DNA

We studied 32 biopsy-matched urine specimens collected from 26 kidney allograft recipients who were enrolled. The study was approved by the Weill Cornell Medicine Institutional Review Board protocols 1207012730). All patients provided written informed consent. Kidney allograft biopsies were classified as acute rejection (n = 11), acute tubular necrosis (n = 12) and normal histology (n = 9) using the Banff 2017 schema [41] (Table 3 and S2 Table in S1 File). DNA was extracted exclusively from urinary cell pellets and deep targeted DNA sequencing was performed on all samples. Briefly, 50cc of fresh urine was centrifuged at 2,000g for 30 minutes at room temperature and the urinary cell pellet was harvested after removing the supernatant. After washing the urine cell pellet with 1ml PBS, the cells were lysed using 350ul of Buffer RLT from Qiagen® and DNA was isolated from the cell pellet using Allprep DNA/RNA/Protein Mini Kit from Qiagen®. Total DNA was quantified using the NanoDrop™ Spectrophotometer. DNA sequencing was performed as previously described for the pair of healthy individuals. Obtained reads were aligned to the human genome reference hg19 using bwa [33]. We filtered out low quality reads using an in-house Python script. We applied the likelihood function on the 1,000 SNV-set to estimate the recipient-specific DNA fraction. The nonparametric Kruskal-Wallis test was applied to assess the correlation between *observed α* and all the diagnosis phenotypes. Dunn's function was applied to test the pairwise association. R software was used for statistical tests and generating graphs [42].

## Ethnicity estimation for donor and recipient

We combined the *observed α* in the kidney transplant patients with a cost function to predict the genotype of both recipient ($g_i^R$) and donor ($g_i^D$) at each SNV *i*. First, we computed the expected fraction of the alternative to the total allele (reference + alternative) $exp_i$ for all 9 possible combinations of $g_i^R$ and $g_i^D$ (Table 4). The observed fraction of the alternative to total alleles ($obs_i$) at the SNV *i* is defined by:

$$obs_i = \frac{Number\ of\ reads\ carrying\ the\ alternative\ allele}{Total\ number\ of\ reads\ covering\ the\ SNV\ i} \tag{6}$$

Then, we used the cost function to determine $g_i^R$ and $g_i^D$ that minimizes the difference between the 9 expected ($exp_i$) and the observed ($obs_i$) fraction of the alternative to total alleles:

$$Loss\left(\alpha, g_i^R, g_i^D\right) = Min_{o=1}^{o=9}(exp_{io} - obs_i)^2 \tag{7}$$

Once $g_i^R$ and $g_i^D$ were estimated, we performed a partial least square analysis (PLS) using 3 subpopulations from the 1,000 Genomes Project African, East Asian and European

**Table 3. Characteristics of kidney transplant recipients.**

| Recipient Characteristics | Patient Total (N = 26) | Patients with AR (N = 8) | Patients with ATI (N = 10) | Patients with No Pathology (N = 8) |
|---|---|---|---|---|
| Number of Biopsy Associated Urine Specimens | 32 | 12 | 11 | 9 |
| Age, years | | | | |
| Mean (SD) | 45.5 (14) | 46.1 (19) | 43.8 (14) | 47.1 (8) |
| Median | 43 | 43 | 40 | 49 |
| Min, Max | 25, 80 | 25, 80 | 29, 74 | 34, 59 |
| Gender, N (%) | | | | |
| Male | 40 (100%) | 40 (100%) | 40 (100%) | 40 (100%) |
| Race, N (%) | | | | |
| White | 8 (31%) | 3 (38%) | 3 (30%) | 2 (25%) |
| Black | 11 (42%) | 4 (50%) | 3 (30%) | 4 (50%) |
| Hispanic | 3 (11%) | 1 (12%) | 0 (0%) | 2 (25%) |
| Asian | 2 (8%) | 0 (0%) | 2 (20%) | 0(0%) |
| Mixed | 2 (8%) | 0 (0%) | 2 (20%) | 0 (0%) |
| Cause of ESRD, N (%) | | | | |
| Diabetes | 4 (15%) | 1 (13%) | 1 (10%) | 2 (25%) |
| Hypertension | 11 (42%) | 3 (37%) | 5 (50%) | 3 (50%) |
| Glomerulonephritis | 6 (23%) | 3 (37%) | 1 (10%) | 2 (25%) |
| Polycystic Kidney Disease | 3 (12%) | 0 (0%) | 2 (20%) | 1 (11%) |
| Other | 2 (8%) | 1 (13%) | 1 (10%) | 0 (0%) |
| Prior Transplant History, N (%) | 2 (8%) | 1 (13%) | 1 (10%) | 0 (0%) |
| Donor Source, N (%) | | | | |
| Living | 17 (65%) | 4 (50%) | 7 (70%) | 6 (75%) |
| Deceased | 9 (35%) | 4 (50%) | 3 (30%) | 2 (25%) |
| Induction Therapy, N (%) | | | | |
| Antithymocyte globulin | 23 (88%) | 5 (62%) | 10 (100%) | 8 (100%) |
| IL-2 Receptor Antibody | 3 (12%) | 3 (38%) | 0 (0%) | 0 (0%) |
| Steroid Maintenance Therapy, N (%) | 10 (38%) | 4 (50%) | 2 (20%) | 4 (50%) |
| Time since Transplant to Biopsy, Month, mean (SD) | 8.26 (12.32) | 9.79 (11.05) | 1.56 (1.64) | 15.10 (17.20) |
| Biopsy Creatinine | 3.1 (2.4) | 2.9 (1.9) | 4.4 (2.9) | 1.6 (0.25) |
| One Year Post Biopsy Creatinine | 2.2 (1.7) | 2.9 (2.9) | 2.0 (0.6) | 1.8 (0.7) |

**Table 4. Expected fraction of the alternative to total alleles (reference + alternative) as a function of *observed α*.**

| $g_i^R$ | $g_i^D$ | $exp_i$ |
|---|---|---|
| 0 | 0 | 0 |
| 0 | 1 | $\alpha/2$ |
| 0 | 2 | $\alpha$ |
| 1 | 0 | $1 - \alpha/2$ |
| 1 | 1 | $1/2$ |
| 1 | 2 | $1 + \alpha/2$ |
| 2 | 0 | $1 - \alpha$ |
| 2 | 1 | $2 - \alpha/2$ |
| 2 | 2 | 1 |

populations using the mixOmics R package [43] and then predicted the ethnicity of donor and recipient in the real kidney transplant samples. The leave-two-out 1,000-fold cross-validation showed the highest prediction accuracy (81.6%) when using the Yoruba in Nigeria, the Southern Han Chinese and the Toscani in Italy amongst the African, East Asian and European subpopulations (S2 Fig in S1 File). We excluded the American and the South Asian populations because using just 1,000 SNVs is not sufficient to perform a reliable PLS on 5 populations, where the highest cross-validation prediction accuracy was too low at only 54.8% (S2 Fig in S1 File). More relevant, none of the donors or recipients involved in the study belonged to the South Asian or the American populations.

## Results

### In silico simulation of donor-recipient DNA mixtures

To determine the optimal sequencing parameters, we use numerical simulations. The simulation process is based on generating two different SNV-sets, then merging the two sets with a predefined proportion of each set; $\alpha$ from set 1 and $(1-\alpha)$ from set 2, and then applying a likelihood function (Methods) to estimate this proportion (*observed $\alpha$*). Two major parameters affect the estimation of the *observed $\alpha$*: the number of sequenced SNVs ($N$) and the depth of sequencing coverage ($M$). For a range of parameters $N$ = {10, 50, 100, 500, 1,000} and $M$ = {10, 50, 100, 500, 1,000, 5,000, 10,000} and varying $\alpha$ from 0 to 0.5 in steps of 0.01, we repeated the simulation process for each $N$ x $M$ x $\alpha$ combination 1,000 times to obtain an empirical distribution of *observed $\alpha$* (S3 Fig in S1 File).

We computed the maximum error ($\varepsilon$) for each combination $N$ x $M$ over all tested $\alpha$. $\varepsilon$ ranges between 0 (best case where *observed $\alpha$* = tested $\alpha$) and 0.5 (worst case where tested $\alpha$ = 0.5 and *observed $\alpha$* = 0 or tested $\alpha$ = 0 and *observed $\alpha$* = 0.5) (Fig 1). As expected, our

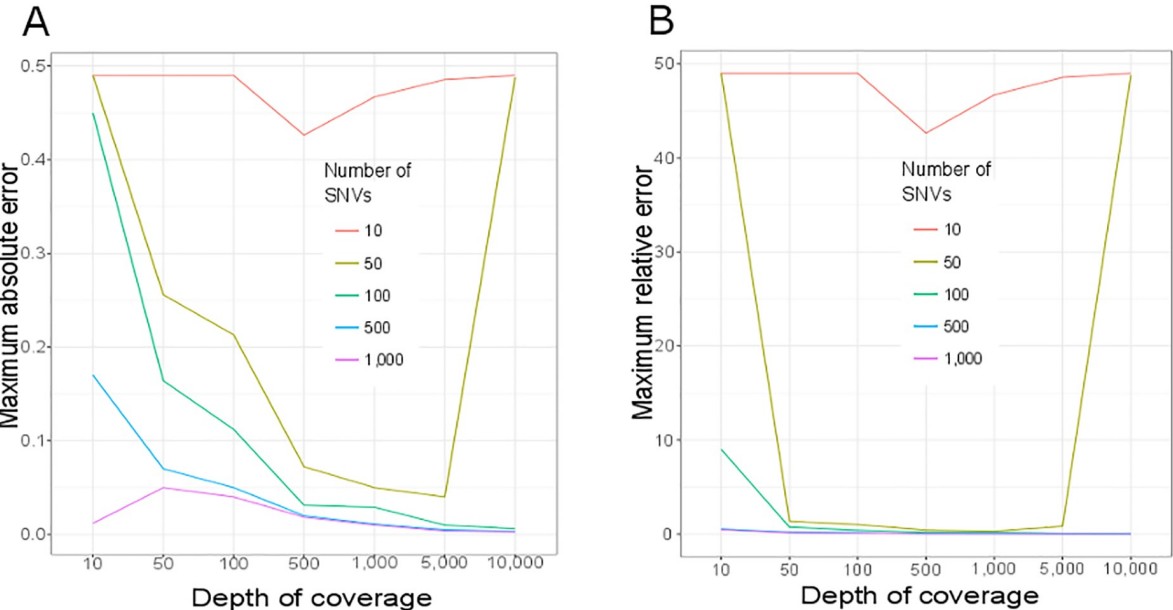

**Fig 1. Maximum error for detecting the DNA fraction $\alpha$ in a simulated DNA sequencing experiment by varying sequencing depth and number of SNVs.** Maximum absolute (A) and maximum relative (B) errors are represented. A total of 35 scenarios combining five different numbers of SNVs $N$ = {10, 50, 100, 500, 1,000} and seven depth of coverage $M$ = {10, 50, 100, 500, 1,000, 5,000, 10,000} were simulated (S3 Fig in S1 File). Represented here are maximum error observed in 1,000 simulations for every tested $\alpha$ ranging from 0 to 0.5 in steps of 0.01.

simulations show that increasing both *N* and *M* improves the *observed α* estimation accuracy. Moreover, the estimation of the *observed α* is unstable when using a small number of SNVs ($N < 100$) or low coverage ($M < 500$). The prediction accuracy stabilizes above $N > 500$ and $M > 1,000$.

## Experimental estimation of α using a controlled mixture of urine from two individuals

To assess the accuracy of detecting *observed α* in a mixture of two real human urine samples, we performed a targeted sequencing of urine DNA from two healthy individuals originating from different populations: S1 a healthy 35-year-old European woman and S2 a healthy 34-year-old Arab woman (S2 Table in S1 File). A total of 1,850 exonic regions from a panel targeting 93 genes known to be associated with risk of breast cancer were sequenced. These sequenced genomic regions cover 370,942 base pairs across 22 chromosomes (S1 Table in S1 File). We chose to use this genomic panel due to its costs and the high number of SNVs present in the targeted genomic regions. Indeed, a total of 51,893 bi-allelic SNVs falling in these genomic regions were present in the Exome Aggregation Consortium (ExAC) [44]. As the method works on bi-allelic SNV with different genotypes between donor and recipient, we computed for each SNV the probability of having different genotypes for two individuals (S1 Table in S1 File). Only 437 SNVs have a probability of having different genotypes for two individuals higher than 10%.

As a measure of quality control, we first checked the balance of reference and alternative alleles in heterozygous calls. The alternative allele frequency is expected to be around 50% in heterozygous genotypes. However, we observed the presence of SNVs with skewed alternative allele frequencies (S4 Fig in S1 File). We noticed the recurrence of such unbalance in every replicate of both samples (S5 Fig in S1 File for examples). We investigated whether the amplification-based strategies for DNA target enrichment affect the allele dropout, causing the skewed alternative allele distribution. We found that the SNVs with a skewed distribution all fall into the primer sequence regions or carried by reads targeted by primers containing SNVs. We therefore filtered out SNVs falling into these regions and kept the 1,000 most common SNVs in the general population. These 1,000 SNVs will be used as a SNV panel for detecting DNA fraction in a combination of two DNA sources (*observed α*) in the rest of the study. The alternative allele frequency was balanced in these 1,000 SNVs (S6 Fig in S1 File). Moreover, the maximum error of estimating the *observed α* based on these 1,000 SNVs in all replicates was $< 0.0034$ (mean error = $0.0028 \pm 0.00037$). We then mixed 90% DNA from S1 and 10% DNA from S2 in three replicates. For each replicate, targeted DNA sequencing was performed and the *observed α* was estimated. The preparation of the mixture was based on total DNA content in the samples. However, the presence of bacterial DNA in urine samples can strongly skew the estimation of human DNA concentration measurement [45]. We assessed the actual DNA concentration of S1 and S2 in urine by considering the mean *observed α* over the three replicates to 0.053. This indicates that S1 DNA concentration is ~19 times lower than S2 DNA concentration. Considering the estimated S1 and S2 DNA concentration, the maximum error of the *observed α* was $< 3.5\%$ in the three replicates (Fig 2).

We extended the analysis to two levels of DNA mixture scenarios: (i) 70% DNA from S1 and 30% DNA from S2, (ii) 50% DNA from S1 and 50% DNA from S2. Each scenario was replicated three times and targeted DNA sequencing was performed for each replicate. The *observed α* was similar in the three replicates of all three scenarios (scenario i: mean *observed α* = $0.11 \pm 0.036$, scenario ii: mean *observed α* = $0.032 \pm 0.00048$). Considering the estimated S1 and S2 DNA concentration, the maximum error of the *observed α* was $< 3.8\%$ in all replicates of both scenarios (0.037 in scenario (i) and 0.018 in scenario (ii)) (Fig 2).

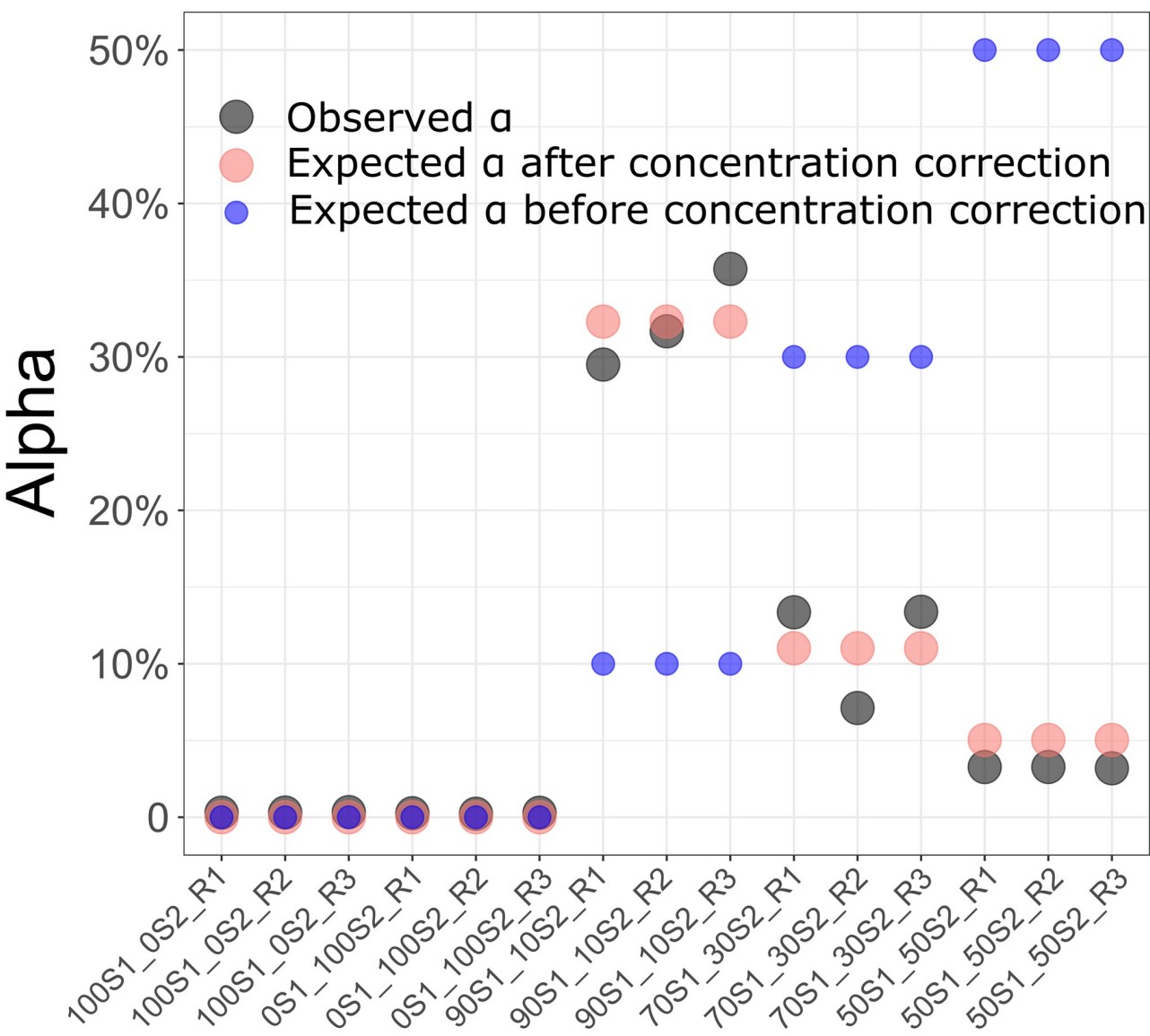

**Fig 2. Estimation of DNA fraction (Alpha) in a combination of two healthy DNA sources.** Five scenarios of DNA mixtures and three replicates for each scenario were performed. From left to right: 100% from individual S1 and 0% from individual S2; 0% from individual S1 and 100% from individual 2; 50% from individual 1 and 50% from individual 2; 70% from individual S1 and 30% from individual S2; 90% from individual S1 and 10% from individual S2. The estimated fractions (*estimated α*) are represented by black dots. The expected fractions when DNA concentration in individual S1 was 19 times lower than DNA concentration in individual S2 are represented by red dots. The expected fractions before correction for DNA concentration are represented by blue dots.

## Simulation of the effect of family relationship and ethnicity on the estimation of α

The most challenging scenario is that of one sibling donating a kidney to another, as they share 50% of their genome. The extreme case of mono-zygotic twins, where both genomes are identical, can of course not be addressed with our method. To numerically explore this "worst case" scenario, we used whole genome sequencing data from 91 siblings [37] and then

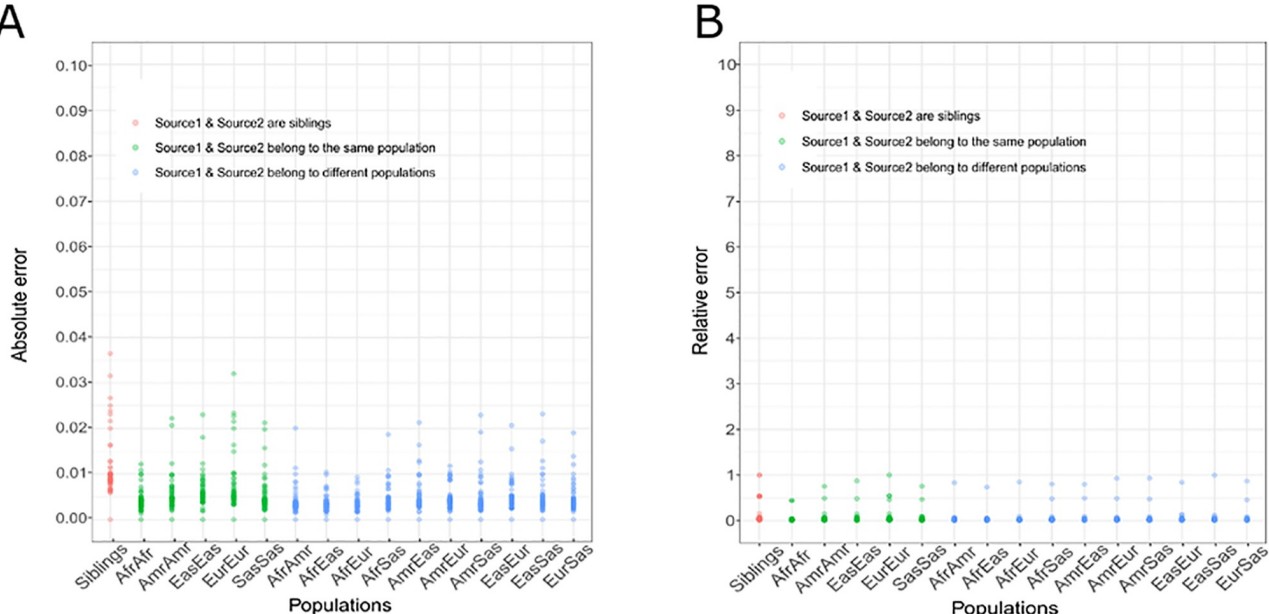

**Fig 3. Effect of family relationship and ethnicity on detecting DNA fraction in a combination of two DNA sources.** Each dot represents in A) the maximum absolute error and in B) the maximum relative error for each expected ($\alpha$) from 0 to 0.5 in steps of 0.01 over 100 pairs of siblings (red), 100 pairs of individuals belonging to the same population (green) and 100 pairs of individuals belonging to different populations (blue). Afr = Africans. Amr = Americans. Eas = East Asians. Eur = Europeans. Sas = South Asians.

generated 100 combinations of every sibling pair. Self-reported relationship was confirmed using the identity by state (S1 Fig in S1 File). For each pair of siblings, we simulated donor and recipient DNA sequences by varying $\alpha$ from 0 to 0.5 in steps of 0.01 and resampling the mean coverage at 5,000 reads. The maximum absolute error was observed when the expected $\alpha$ = 0.07: *observed $\alpha$* = 0.034 (Fig 3).

## Simulation of the effect of population origin on the estimation of $\alpha$

Using the same methods as when comparing siblings, we then assessed the effect of donor and recipient ethnicity on our method. We applied our method to simulated pairs of individuals belonging to the same and to different populations of the 1,000 genomes project [36]: Africans, Americans, East Asians, Europeans and South Asians. The absolute error was < 0.04 in all scenarios (Fig 3). As expected, the absolute error was lower when the two DNA sources belonged to different populations (mean maximum absolute error = 0.018 ± 0.005) than when they belonged to the same population (mean maximum absolute error = 0.022 ± 0.007). Additionally, the maximum relative error was comparable in all scenarios (mean maximum relative error = 0.836 ± 0.137) whether the two DNA sources belonged to the same (mean maximum relative error = 0.764 ± 0.207) or different populations (mean maximum relative error = 0.856 ± 0.077). These results confirm the power of the method for detecting the DNA fraction in a combination of two DNA sources independent of familial relationship or ethnicity.

## Application to urine samples from clinical kidney allograft patients

To test our method in a real-case scenario, we used DNA extracted from 32 urine samples matched to 32 biopsies from 26 kidney allograft recipients and classified the urine samples

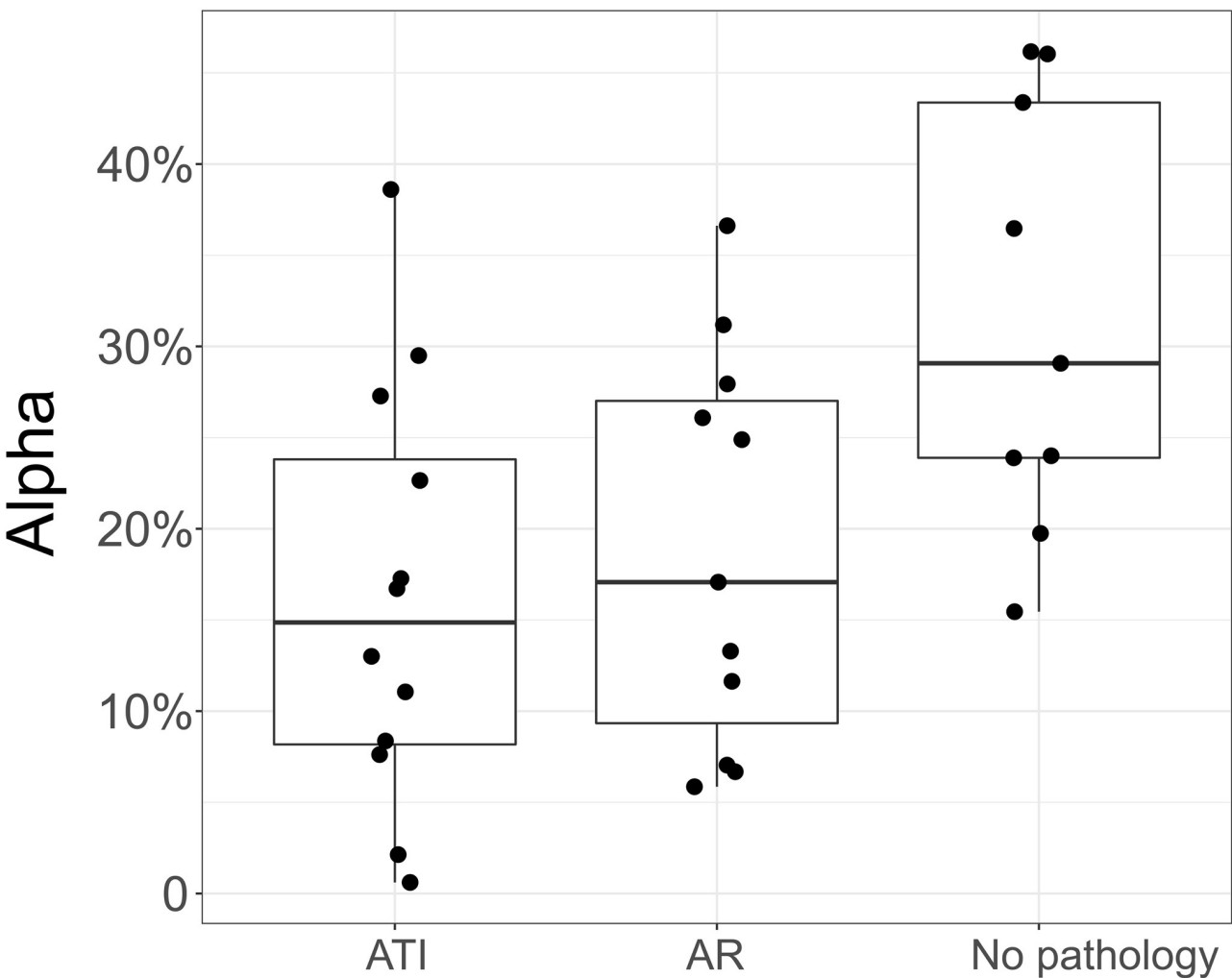

**Fig 4. Donor/recipient to total DNA fraction in urine from 32 real kidney allograft recipients.** Box plots and individual data points of the estimated fraction (*observed* $\alpha$) are estimated from deep DNA targeted sequencing of urinary cells. AR: Acute Rejection. ATI: Acute Tubular Injury. A statistically significant difference was observed between all the diagnostic categories (P = 0.035, Kruskal-Wallis test). By Dunn's test, difference in *observed* $\alpha$ between the two pathologies and no pathology group was statistically significant: ATI vs no-pathology: P = 0.0064 and AR vs no-pathology: P = 0.026. The pairwise comparison of AR and ATI pathologies was not statistically significant (P > 0.05).

into three groups based on their Banff classification of kidney alllograft biopsy: "Acute Tubular Injury" (ATI, N = 12), "Acute Rejection" (AR, N = 11) and "No Observed Pathology" (N = 9) (Table 3 and S2 Table in S1 File). DNA was extracted from the urinary cells and deep targeted sequencing was performed for the 32 samples. Reflecting the effect of depth of coverage on the accuracy of detecting *observed* $\alpha$ in simulated data, we set the mean depth of coverage to ~ 14,000 reads. After read alignment, we applied our method to estimate the donor/recipient to total DNA fraction (Fig 4).

The difference of *observed* $\alpha$ between the diagnosis phenotypes was statistically significant (P = 0.035, Kruskal-Wallis test). We observed a significant difference when comparing the two transplant kidney pathologies ATI and AR to the No Pathology group (P = 0.0064 and P = 0.026, Dunn's test for ATI vs no pathology and AR vs no pathology, respectively). However, no significant difference was observed in *observed* $\alpha$ when comparing the two pathologies ATI to AR (P = 0.31, Dunn's test).

### Inference of donor and recipient ethnic origin

In the absence of donor and recipient genomes, it is impossible to determine whether the *observed α* represents the donor or the recipient fraction of the total DNA. However, in cases in which recipient and donor gender or ethnicity differ, this issue can be addressed. The urinary cell DNA sequencing we performed here did not target genomic regions of the Y chromosome. Thus, detecting recipient and donor gender cannot be carried out using the actual data, but could be accomplished/carried out easily in future sequencing panels.

To predict donor and recipient ethnicity, an estimation of both recipient and donor genotypes is needed. For each of 1,000 SNVs, we computed the fraction of the alternative to total alleles. We then used the *observed α* to compute the nine expected fractions of the alternative allele (Table 4). We then used a cost function to estimate donor and recipient genotypes that minimizes the difference between the nine expected fractions and the observed fraction of the alternative to total alleles. Based on these estimated genotypes, we applied a supervised classification method to predict the recipient and the donor ethnicity as following: as donor-specific DNA fraction has been shown to be higher in the no-pathology group [28], we supposed the *observed α* to represent the donor-to-total DNA fraction and computed the probability of donor and recipient belonging to one of the three populations: African, East Asian and European (see Methods). Both donor and recipient are assigned to the population showing the highest probability and then compared to the self-reported ethnicity. Seven recipients and eight donors were excluded from the prediction because they belong to a mixed self-reported population or the *observed α* was ~ 0 so the prediction of donor genotypes was impossible. The prediction was inconclusive (Probability of prediction < 70%) for 5 recipients and 8 donors. In 16 of 20 recipients (80%) and 15 of 16 donors (94%), the probability of prediction was higher than 70%. However, only one AR sample and one ATI sample, for which the prediction was conclusive, had donor and recipient ethnicity mismatch. In these two samples (European donor and African recipient for both samples), the prediction was in agreement with the self-reported ethnicity. Hence, due to the small number of self-reported ethnicity mismatches, it is impossible to confirm whether the *observed α* represents the donor or the recipient DNA fraction (as *observed α* < 0.5 by definition).

## Discussion

Different omics technologies, including mRNA measurement by PCR [7], metabolomics [12] and RNA-sequencing [13] have been applied by our group and others to identify non-invasive biomarkers for kidney allograft rejection. Here, we present a new approach based on targeted deep- sequencing of DNA obtained from urine samples from kidney allograft recipients. We extended methods originally used for the assessment of DNA contamination to estimate the fraction of recipient DNA in a two-source mixed DNA sample [29, 30]. We used in silico simulations to obtain a suitable parameter range for the method to be sufficiently accurate in estimating the fraction of a two-source DNA mixture. We then experimentally evaluated the accuracy of the estimation method using controlled mixtures of two DNA sources. Allele drop-out occurs in amplification-based target enrichment when a variant is located in a primer region and prevents primer hybridization, leading to failed amplification and allele bias [34]. Our method overcomes these unexpected artefacts due to DNA sequencing. Other algorithms for estimating the donor-specific DNA fraction require the donor and recipient relationship information [22]. Here, we found that ethnicity and familial relationship between donor and recipient appear to have a lower impact as compared to previously presented methods.

We tested our method on clinical samples from patients with and without kidney allograft rejection events. We compared the α value obtained from urine DNA sequencing reads of

kidney allograft recipients with kidney injury associated with AR and ATI. The alpha value was significantly different in patients with AR and ATI compared to those without kidney allograft pathology. The calculation of alpha is based on the assumption that the DNA isolated from the urine is derived from the transplanted kidney and that both the recipient and the donor DNA are present: recipient DNA from the infiltrating immune cells and donor DNA from the kidney parenchymal cells. Indeed, by counting Y chromosome-derived cell free DNA, we have recently shown that in kidney recipients with donor-recipient gender mismatch the donor-specific DNA fraction was lower in recipients with UTI compared to those with no UTI, and higher in recipients with BKVN compared to those with no BKVN [28]. Thus, our approach might be considered as a potential new diagnostic signature measured in urine specimens.

We were not able to ascertain whether the DNA in recipient urine is derived mostly from the donor or the recipient. Studies have shown that both AR and ATI are associated with allograft damage indicating that there will be some donor DNA in the urine. But AR is also associated with recipient immune cell infiltration while ATI is usually not [46]. Tissue injury from ATI however could be associated inflammation. The fraction of recipient to donor cells in the urine should be higher for AR compared to ATI and the fraction of donor cell to recipient cells in the urine should be higher for ATI compared to AR. Thus, AR patients should have a fraction of donor to recipient DNA of much lower than 0.5 and ATI patients should have a fraction of donor to recipient DNA of much greater than 0.5. To address this, a future complementary analysis on a bigger sample having donor and recipient ethnicity and/or gender mismatches will be worthy of investigation.

## Supporting information

**S1 File.**
(ZIP)

## Author Contributions

**Conceptualization:** Manikkam Suthanthiran, Karsten Suhre.

**Data curation:** Aziz Belkadi, Gaurav Thareja, Darshana Dadhania.

**Formal analysis:** Aziz Belkadi, Gaurav Thareja.

**Funding acquisition:** Aziz Belkadi, Manikkam Suthanthiran, Karsten Suhre.

**Investigation:** Aziz Belkadi, Manikkam Suthanthiran, Karsten Suhre.

**Project administration:** Aziz Belkadi, Manikkam Suthanthiran, Karsten Suhre.

**Resources:** Gaurav Thareja, John R. Lee, Thangamani Muthukumar, Catherine Snopkowski, Carol Li, Anna Halama, Sara Abdelkader, Silvana Abdulla, Yasmin Mahmoud, Joel Malek, Manikkam Suthanthiran, Karsten Suhre.

**Software:** Aziz Belkadi, Karsten Suhre.

**Supervision:** Manikkam Suthanthiran, Karsten Suhre.

**Validation:** Aziz Belkadi, Darshana Dadhania, Manikkam Suthanthiran, Karsten Suhre.

**Visualization:** Aziz Belkadi.

**Writing – original draft:** Aziz Belkadi, Darshana Dadhania.

**Writing – review & editing:** Aziz Belkadi, Manikkam Suthanthiran.

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
