## [Decision Letter · Decision Letter 0]

11 Feb 2021

PONE-D-20-40848

Deep sequencing of DNA from urine of kidney allograft recipients to estimate the donor-specific DNA fraction

PLOS ONE

Dear Dr. Belkadi,

Thank you for submitting your manuscript to PLOS ONE. After careful consideration, we feel that it has merit but does not fully meet PLOS ONE’s publication criteria as it currently stands. Therefore, we invite you to submit a revised version of the manuscript that addresses the points raised during the review process.

he authors need to address all reviewer's comments:

Reviewer # 1:

The authors have a creative concept that the degree of kidney allograft rejection can be estimated by deep sequencing DNA found in recipient urine.  An analysis of the SNV frequencies is then proposed as a means of predicting the relative contributions of donor versus recipient DNA; the hypothesis is that, with increasing recipient DNA contribution, one can infer an increased proportion of responding immune cells and therefore a higher likelihood of allograft rejection.

The concept seems promising, but I fear that the authors gloss over some significant caveats to this method. The first is the assumption that recipient-specific DNA originates mostly from tissue-invading immune cells; as the authors note, they were able to previously perform such an analysis on kidney recipients with urinary tract infections and showed lower proportions of donor DNA when individuals were diagnosed with UTI (compared to individuals without). This does seem to indicate that recipient-specific DNA can come from recipient immune cells responding to a urinary tract infection. In a clinical setting, do you anticipate being able to tell the difference between a low donor-specific fraction (as a result of low rejection) and a low donor-specific fraction (as a result of competition by infiltrating immune cells due to infection)?  This would be particularly relevant for subclinical UTIs.

Secondly, I would caution the authors against stating that they can determine the donor-specific DNA fraction, given that this method is unable to determine which DNA specimen is from which individual. I agree with their interpretation that the fraction of recipient to donor cells SHOULD be higher for AR than for ATI (and for no-pathology specimens), however, as they state, that future complementary study would be necessary.

Specific questions about the procedure:

1)     Could the authors address why they chose to use the Breast cancer risk panel rather than other available genomic sequencing panels?

2)     Does this method use cell free DNA, or does it include a cell lysis step? Are the authors concerned about the relative proportions of donor/recipient DNA in the cell free vs. cellular fractions?

3)     Regarding DNA target enrichment due to skewed amplification, I am curious about the SNVs that are NOT in the primer regions. I presume that, if one of the amplification primers fails to bind effectively due to a SNV, it affects the results not just of that SNV but of all SNVs within that amplicon? I understand and wholeheartedly agree with the authors’ decision to exclude SNVs falling within primer sequence regions; I wonder if the quantification of other SNVs is negatively affected by these binding issues as well.

I appreciate the commitment to patient privacy, and do not expect the authors to share specific point mutations for their study subjects.  However, the manuscript would benefit from the inclusion of some additional data, for example, concentrations of extracted DNA both before and after library preparation, specific numbers for the donor-specific fraction of each tested patient, etc.

Overall, the authors should be commended on a well-written paper that clearly expresses their concepts and logical process. Some sentences are awkwardly written and could benefit from a second look, but it doesn’t reach the level of incomprehension

We look forward to receiving your revised manuscript.

Kind regards,

Stanislaw Stepkowski

Academic Editor

PLOS ONE

Journal Requirements:

2) Your ethics statement should only appear in the Methods section of your manuscript. If your ethics statement is written in any section besides the Methods, please move it to the Methods section and delete it from any other section. Please ensure that your ethics statement is included in your manuscript, as the ethics statement entered into the online submission form will not be published alongside your manuscript.

3) In line with PLOS' guidelines on detailed reporting (https://journals.plos.org/plosone/s/criteria-for-publication#loc-3), please ensure you have provided sufficient detail on participant recruitment.

4) Please improve statistical reporting and report exact p-values for all values greater than or equal to 0.001. P-values less than 0.001 should be expressed as p < 0.001. Our statistical reporting guidelines are available at https://journals.plos.org/plosone/s/submission-guidelines#loc-statistical-reporting.

5)  We note that the grant information you provided in the ‘Funding Information’ and ‘Financial Disclosure’ sections do not match.

6) We note that you have indicated that data from this study are available upon request. PLOS only allows data to be available upon request if there are legal or ethical restrictions on sharing data publicly. For information on unacceptable data access restrictions, please see http://journals.plos.org/plosone/s/data-availability#loc-unacceptable-data-access-restrictions.

7) Please include captions for your Supporting Information files at the end of your manuscript, and update any in-text citations to match accordingly. Please see our Supporting Information guidelines for more information: http://journals.plos.org/plosone/s/supporting-information.

Reviewers' comments:

Reviewer's Responses to Questions

**Comments to the Author**

1. Is the manuscript technically sound, and do the data support the conclusions?

Reviewer #1: Partly

Reviewer #2: Yes

2. Has the statistical analysis been performed appropriately and rigorously? 

Reviewer #1: I Don't Know

Reviewer #2: I Don't Know

3. Have the authors made all data underlying the findings in their manuscript fully available?

Reviewer #1: No

Reviewer #2: Yes

4. Is the manuscript presented in an intelligible fashion and written in standard English?

Reviewer #1: Yes

Reviewer #2: Yes

5. Review Comments to the Author

Reviewer #1: The authors have a creative concept that the degree of kidney allograft rejection can be estimated by deep sequencing DNA found in recipient urine. An analysis of the SNV frequencies is then proposed as a means of predicting the relative contributions of donor versus recipient DNA; the hypothesis is that, with increasing recipient DNA contribution, one can infer an increased proportion of responding immune cells and therefore a higher likelihood of allograft rejection.

The concept seems promising, but I fear that the authors gloss over some significant caveats to this method. The first is the assumption that recipient-specific DNA originates mostly from tissue-invading immune cells; as the authors note, they were able to previously perform such an analysis on kidney recipients with urinary tract infections and showed lower proportions of donor DNA when individuals were diagnosed with UTI (compared to individuals without). This does seem to indicate that recipient-specific DNA can come from recipient immune cells responding to a urinary tract infection. In a clinical setting, do you anticipate being able to tell the difference between a low donor-specific fraction (as a result of low rejection) and a low donor-specific fraction (as a result of competition by infiltrating immune cells due to infection)? This would be particularly relevant for subclinical UTIs.

Secondly, I would caution the authors against stating that they can determine the donor-specific DNA fraction, given that this method is unable to determine which DNA specimen is from which individual. I agree with their interpretation that the fraction of recipient to donor cells SHOULD be higher for AR than for ATI (and for no-pathology specimens), however, as they state, that future complementary study would be necessary.

Specific questions about the procedure:

1) Could the authors address why they chose to use the Breast cancer risk panel rather than other available genomic sequencing panels?

2) Does this method use cell free DNA, or does it include a cell lysis step? Are the authors concerned about the relative proportions of donor/recipient DNA in the cell free vs. cellular fractions?

3) Regarding DNA target enrichment due to skewed amplification, I am curious about the SNVs that are NOT in the primer regions. I presume that, if one of the amplification primers fails to bind effectively due to a SNV, it affects the results not just of that SNV but of all SNVs within that amplicon? I understand and wholeheartedly agree with the authors’ decision to exclude SNVs falling within primer sequence regions; I wonder if the quantification of other SNVs is negatively affected by these binding issues as well.

I appreciate the commitment to patient privacy, and do not expect the authors to share specific point mutations for their study subjects. However, the manuscript would benefit from the inclusion of some additional data, for example, concentrations of extracted DNA both before and after library preparation, specific numbers for the donor-specific fraction of each tested patient, etc.

Overall, the authors should be commended on a well-written paper that clearly expresses their concepts and logical process. Some sentences are awkwardly written and could benefit from a second look, but it doesn’t reach the level of incomprehension.

Reviewer #2: I'm really sorry,but I must say that I'm not competent to review this paper despite the the fact that researchfield is my field (kidney) and the paper seems very interesting (that's why I accepted to be reviewer), but when I read the methods and results parts, there is too much informatic and mathematic for me.

6. PLOS authors have the option to publish the peer review history of their article (what does this mean?). If published, this will include your full peer review and any attached files.

Reviewer #1: No

Reviewer #2: No

---

## [Author Response · Author response to Decision Letter 0]

23 Mar 2021

Comments from Reviewer #1

Comment 0-1: The first is the assumption that recipient-specific DNA originates mostly from tissue-invading immune cells; as the authors note, they were able to previously perform such an analysis on kidney recipients with urinary tract infections and showed lower proportions of donor DNA when individuals were diagnosed with UTI (compared to individuals without). This does seem to indicate that recipient-specific DNA can come from recipient immune cells responding to a urinary tract infection. In a clinical setting, do you anticipate being able to tell the difference between a low donor-specific fraction (as a result of low rejection) and a low donor-specific fraction (as a result of competition by infiltrating immune cells due to infection)? This would be particularly relevant for subclinical UTIs.

Response: Agree. Thank you for this interesting comment. In a clinical situation like UTI, immune cells detected in urine could indeed be originated from recipient’s reaction to the infection. Based on the present observations only, we cannot state the origin of a low donor-specific DNA fraction, especially in UTI patients. Because we did not recruit patients with UTI in this study, and deconvolution methods -like our previous paper mentioned by the reviewer- to estimate cell-type composition work on RNA-seq or methylation but not on DNA-seq data. Further investigations on more phenotypes including UTI and using state-of-the art single cell RNA seq will help to address the reviewer’s insightful comment.

Comment 0-2: Secondly, I would caution the authors against stating that they can determine the donor-specific DNA fraction, given that this method is unable to determine which DNA specimen is from which individual. I agree with their interpretation that the fraction of recipient to donor cells SHOULD be higher for AR than for ATI (and for no-pathology specimens), however, as they state, that future complementary study would be necessary.

Response: We thank the reviewer for the important point. `we have updated the manuscript by changing “donor-specific” to donor/recipient specific”

Comment 0-3: I appreciate the commitment to patient privacy, and do not expect the authors to share specific point mutations for their study subjects. However, the manuscript would benefit from the inclusion of some additional data, for example, concentrations of extracted DNA both before and after library preparation, specific numbers for the donor-specific fraction of each tested patient, etc.

Response: Thank you for pointing this out. We added a Supplementary Table 2 showing the DNA concentration and the observed alpha for each sample included in the analysis.

Comment 0-4: Overall, the authors should be commended on a well-written paper that clearly expresses their concepts and logical process. Some sentences are awkwardly written and could benefit from a second look, but it doesn’t reach the level of Incomprehension

Response: Thank you for your comment. The manuscript has been edited by an expert scientific writer.

Comment 1: Could the authors address why they chose to use the Breast cancer risk panel rather than other available genomic sequencing panels?

Response: Thank you for pointing this out. Our idea was to estimate donor (recipient) – specific DNA fraction in recipient urine. Our hypothesis was that any large targeted genomic regions will present different alleles between donor and recipients (except for monozygotic twins) that could be used to compute the likelihood function. The breast cancer risk panel covered a large genomic region that display this characteristic. Additionally, the costs of this panel corroborated our choice. Furthermore, we have extensive experience working on the breast cancer risk panel in our lab. Nonetheless, we agree with the reviewer that any other DNA-sequencing panel targeting sufficient large and variable (rich in single nucleotide variants) genomic regions will probably give similar results. We have updated the manuscript according to the reviewer’s comment.

Comment 2: Does this method use cell free DNA, or does it include a cell lysis step? Are the authors concerned about the relative proportions of donor/recipient DNA in the cell free vs. cellular fractions?

Response: We thank the reviewer for this comment. We included in our analysis only DNA coming from whole urinary pallets. The pellet DNA has a higher quality as compared to fragmented DNA. For example, the urinary cell-free DNA has a shorter half-life than blood cell-free DNA [27317895]. Additionally, the loss of urinary cell-free DNA often occurs [32325682]. Despite the various commercial stabilization and preservation solutions that were developed and used for urinary cell-free DNA applications [33207777], studies evaluating the efficacy of these commercial products are largely missing. Finally, it has been shown that urinary single-cell DNA needs quasi-immediate sample storage conditions and suitable preservative to avoid cell lysis and DNA degradation [32251424]. For all these reasons, we decided to run urinary pellet DNA sequencing. We agree with the reviewer that it would be interesting to estimate donor (recipient) -specific DNA fraction in urine cell-free DNA and compare it to urinary pellet DNA. But this will need further investigations. We have updated the manuscript according to the reviewer’s comment. 

Comment 3: Regarding DNA target enrichment due to skewed amplification, I am curious about the SNVs that are NOT in the primer regions. I presume that, if one of the amplification primers fails to bind effectively due to a SNV, it affects the results not just of that SNV but of all SNVs within that amplicon? I understand and wholeheartedly agree with the authors’ decision to exclude SNVs falling within primer sequence regions; I wonder if the quantification of other SNVs is negatively affected by these binding issues as well.

Response: Thank you for highlighting this point. In the final list of 1,000 variants used in our analysis, we filtered out the variants falling in the primer sequences +/- the read length (150 bp). We are sorry for the misunderstanding. As requested by the reviewer, we checked the alternative allele frequency for single nucleotide variants falling in primer sequences and the reads targeted by a primer containing a variant. We identified 6,979 SNVs falling in primer sequences and 17,258 SNVs carried by reads targeted by primers containing SNVs. We filtered out uncovered SNVs in each replicate. For both variants falling in primer sequences (Fig A) and variants carried by reads targeted by a primer containing a variant (Fig B), the alternative allele distribution was skewed. This observation supports the reviewer’s hypothesis. We have updated the manuscript according to the reviewer’s comment.

---

## [Editor Report · Decision Letter 1]

29 Mar 2021

Deep sequencing of DNA from urine of kidney allograft recipients to estimate  donor/recipient-specific DNA fractions

PONE-D-20-40848R1

Dear Dr. Belkadi,

We’re pleased to inform you that your manuscript has been judged scientifically suitable for publication and will be formally accepted for publication once it meets all outstanding technical requirements.

Kind regards,

Stanislaw Stepkowski

Academic Editor

PLOS ONE
---

## [Editor Report · Acceptance letter]

6 Apr 2021

PONE-D-20-40848R1 

Deep sequencing of DNA from urine of kidney allograft recipients to estimate  donor/recipient-specific DNA fractions 

Dear Dr. Suhre:

I'm pleased to inform you that your manuscript has been deemed suitable for publication in PLOS ONE. Congratulations! Your manuscript is now with our production department. 

Kind regards, 

on behalf of

Dr. Stanislaw Stepkowski 

Academic Editor

PLOS ONE